# Multitask learning meets tensor factorization: task imputation via convex optimization

**Kishan Wimalawarne**
Tokyo Institute of Technology
Meguro-ku, Tokyo, Japan
kishan@sg.cs.titech.ac.jp

**Masashi Sugiyama**
The University of Tokyo
Bunkyo-ku, Tokyo, Japan
sugi@k.u-tokyo.ac.jp

**Ryota Tomioka**
TTI-C
Illinois, Chicago, USA
tomioka@ttic.edu

## Abstract

We study a multitask learning problem in which each task is parametrized by a weight vector and indexed by a pair of indices, which can be e.g, (consumer, time). The weight vectors can be collected into a tensor and the (multilinear-)rank of the tensor controls the amount of sharing of information among tasks. Two types of convex relaxations have recently been proposed for the tensor multilinear rank. However, we argue that both of them are not optimal in the context of multitask learning in which the dimensions or multilinear rank are typically heterogeneous. We propose a new norm, which we call the scaled latent trace norm and analyze the excess risk of all the three norms. The results apply to various settings including matrix and tensor completion, multitask learning, and multilinear multitask learning. Both the theory and experiments support the advantage of the new norm when the tensor is not equal-sized and we do not a priori know which mode is low rank.

## 1 Introduction

We consider supervised multitask learning problems [1, 6, 7] in which the tasks are indexed by a pair of indices known as multilinear multitask learning (MLMTL) [17, 19]. For example, when we would like to predict the ratings of different aspects (e.g., quality of service, food, etc) of restaurants by different customers, the tasks would be indexed by *aspects* × *customers*. When each task is parametrized by a weight vector over features, the goal would be to learn a *features* × *aspects* × *customers* tensor. Another possible task dimension would be *time*, since the ratings may change over time.

This setting is interesting, because it would allow us to exploit the similarities across different customers as well as similarities across different aspects or time-points. Furthermore this would allow us to perform *task imputation*, that is to learn weights for tasks for which we have no training examples. On the other hand, the conventional matrix-based multitask learning (MTL) [2, 3, 13, 16] may fail to capture the higher order structure if we consider learning a flat *features* × *tasks* matrix and would require at least $r$ samples, where $r$ is the rank of the matrix to be learned, for each task.

Recently several norms that induce low-rank tensors in the sense of Tucker decomposition or multilinear singular value decomposition [8, 9, 14, 25] have been proposed. The mean squared error for recovering a $n_1 \times \cdots \times n_K$ tensor of multilinear rank $(r_1, \ldots, r_K)$ from its noisy version scale as $O((\frac{1}{K} \sum_{k=1}^{K} \sqrt{r_k})^2 (\frac{1}{K} \sum_{k=1}^{K} 1/\sqrt{n_k})^2)$ for the *overlapped trace norm* [23]. On the other hand, the error of the *latent trace norm* scales as $O(\min_k r_k / \min_k n_k)$ in the same setting [21]. Thus while the latent trace norm has the better dependence in terms of the multilinear rank $r_k$, it has the worse dependence in terms of the dimensions $n_k$.

Tensors that arise in multitask learning typically have heterogeneous dimensions. For example, the number of aspects for a restaurant (quality of service, food, atmosphere, etc.) would be much

Table 1: Tensor denoising performance using different norms. The mean squared error $\|\hat{\mathcal{W}} - \mathcal{W}^*\|_F^2/N$ is shown for the denoising algorithms (3) using different norms for tensors.

| Overlapped trace norm | Latent trace norm | Scaled latent trace norm |
|---|---|---|
| $O_p\left(\left(\frac{1}{K}\sum_{k=1}^{K}\sqrt{r_k}\right)^2\left(\frac{1}{K}\sum_{k=1}^{K}1/\sqrt{n_k}\right)^2\right)$ | $O_p\left(\min_k r_k / \min_k n_k\right)$ | $\boldsymbol{O_p\left(\min_k\left(r_k/n_k\right)\right)}$ |

smaller than the number of customers or the number of features. In addition, it is a priori unclear which mode (or dimension) would have the most redundancy or sharing that could be exploited by multitask learning. Some of the modes may have full ranks if there is no sharing of information along them. Therefore, both the latent trace norm and the overlapped trace norm would suffer either from the heterogeneous multilinear rank or the heterogeneous dimensions in this context.

In this paper, we propose a modification to the latent trace norm whose mean squared error scales as $O(\min_k(r_k/n_k))$ in the same setting, which is better than both the previously proposed extensions of trace norm for tensors. We study the excess risk of the three norms through their Rademacher complexities in various settings including matrix completion, multitask learning, and MLMTL. The new analysis allows us to also study the tensor completion setting, which was only empirically studied in [22, 23]. Our analysis consistently shows the advantage of the proposed scaled latent trace norm in various settings in which the dimensions or ranks are heterogeneous. Experiments on both synthetic and real data sets are also consistent with our theoretical findings.

## 2  Norms for tensors and their denoising performance

Let $\mathcal{W} \in \mathbb{R}^{n_1 \times \cdots \times n_K}$ be a $K$-way tensor. We denote the total number of entries by $N := \prod_{k=1}^{K} n_k$. A mode-$k$ fiber of $\mathcal{W}$ is an $n_k$ dimensional vector we obtain by fixing all but the $k$th index. The mode-$k$ unfolding $\boldsymbol{W}_{(k)}$ of $\mathcal{W}$ is the $n_k \times N/n_k$ matrix formed by concatenating all the $N/n_k$ mode-$k$ fibers along columns. We say that $\mathcal{W}$ has multilinear rank $(r_1, \ldots, r_K)$ if $r_k = \text{rank}(\boldsymbol{W}_{(k)})$.

### 2.1  Existing norms for tensors

First we review two norms proposed in literature in order to convexify tensor decomposition.

The overlapped trace norm (see [12, 15, 18, 22]) is defined as the sum of the trace norms of the mode-$k$ unfoldings as follows:

$$\|\mathcal{W}\|_{\text{overlap}} = \sum_{k=1}^{K} \|\boldsymbol{W}_{(k)}\|_{\text{tr}}, \tag{1}$$

where $\|\cdot\|_{\text{tr}}$ is the trace norm (also known as the nuclear norm) [10, 20], which is defined as the absolute sum of singular values. Romera-Paredes et al. [17] has used the overlapped trace norm in MLMTL.

The latent trace norm [21, 22] is defined as the infimum over $K$ tensors as follows:

$$\|\mathcal{W}\|_{\text{latent}} = \inf_{\mathcal{W}^{(1)}+\cdots+\mathcal{W}^{(K)}=\mathcal{W}} \sum_{k=1}^{K} \|\boldsymbol{W}_{(k)}^{(k)}\|_{\text{tr}}. \tag{2}$$

Table 1 summarizes the denoising performance in mean squared error analyzed in Tomioka and Suzuki [21] for the above two norms. The setting is as follows: we observe a noisy version $\mathcal{Y}$ of a tensor $\mathcal{W}^*$ with multilinear rank $(r_1, \ldots, r_K)$ and would like to recover $\mathcal{W}^*$ by solving

$$\hat{\mathcal{W}} = \underset{\mathcal{W}}{\text{argmin}} \left(\frac{1}{2}\|\mathcal{W}-\mathcal{Y}\|_F^2 + \lambda \|\mathcal{W}\|_\star\right), \tag{3}$$

where $\|\cdot\|_\star$ is either the overlapped trace norm or the latent trace norm. We can see that while the latent trace norm has the better dependence in terms of the multilinear rank, it has the worse dependence in terms of the dimensions. Intuitively, the latent trace norm recognizes the mode with the lowest rank. However, it does not have a good control of the dimensions; in fact, the factor

$1/\min_k n_k$ comes from the fact that for a random tensor $\mathcal{X}$ with i.i.d. Gaussian entries, the expectation of the dual norm $\|\mathcal{X}\|_{\text{latent}^*} = \max_k \|\boldsymbol{X}_{(k)}\|_{\text{op}}$ behaves like $O_p(\sqrt{\max_k N/n_k})$, where $\|\cdot\|_{\text{op}}$ is the operator norm.

## 2.2   A new norm

In order to correct the unfavorable behavior of the dual norm, we propose the *scaled latent trace norm*. It is defined similarly to the latent trace norm with weights $1/\sqrt{n_k}$ as follows:

$$\|\mathcal{W}\|_{\text{scaled}} = \inf_{\mathcal{W}^{(1)}+\cdots+\mathcal{W}^{(K)}=\mathcal{W}} \sum_{k=1}^{K} \frac{1}{\sqrt{n_k}}\|\boldsymbol{W}^{(k)}_{(k)}\|_{\text{tr}}. \tag{4}$$

Now the expectation of the dual norm $\|\mathcal{X}\|_{\text{scaled}^*} = \max_k \sqrt{n_k}\|\boldsymbol{X}_{(k)}\|_{\text{op}}$ behaves like $O_p(\sqrt{N})$ for $\mathcal{X}$ with random i.i.d. Gaussian entries and combined with the following relation

$$\|\mathcal{W}\|_{\text{scaled}} \leq \min_k \sqrt{\frac{r_k}{n_k}}\,\|\mathcal{W}\|_F\,, \tag{5}$$

we obtain the scaling of the mean squared error in the last column of Table 1. We can see that the scaled latent norm recognizes the mode with the lowest rank *relative to its dimension*.

## 3   Theory for multilinear multitask learning

We consider $T = PQ$ supervised learning tasks. Training samples $(\boldsymbol{x}_{ipq}, y_{ipq})_{i=1}^{m_{pq}}$ $((p,q) \in S)$ are provided for a relatively small fraction of the task index pairs $S \subset [P] \times [Q]$. Each task is parametrized by a weight vector $\boldsymbol{w}_{pq} \in \mathbb{R}^d$, which can be collected into a 3-way tensor $\mathcal{W} = (\boldsymbol{w}_{pq}) \in \mathbb{R}^{d \times P \times Q}$ whose $(p,q)$ fiber is $\boldsymbol{w}_{pq}$. We define the learning problem as follows:

$$\hat{\mathcal{W}} = \operatorname*{argmin}_{\mathcal{W}\in\mathbb{R}^{d\times P\times Q}} \hat{L}(\mathcal{W}), \quad \text{subject to} \quad \|\mathcal{W}\|_{\star} \leq B_0, \tag{6}$$

where the norm $\|\cdot\|_{\star}$ is either the overlapped trace norm, latent trace norm, or the scaled latent trace norm, and the empirical risk $\hat{L}$ is defined as follows:

$$\hat{L}(\mathcal{W}) = \frac{1}{|S|} \sum_{(p,q)\in S} \frac{1}{m_{pq}} \sum_{i=1}^{m_{pq}} \ell\left(\langle \boldsymbol{x}_{ipq}, \boldsymbol{w}_{pq}\rangle - y_{ipq}\right).$$

The true risk we are interested in minimizing is defined as follows:

$$L(\mathcal{W}) = \frac{1}{PQ} \sum_{p,q} \mathbb{E}_{(\boldsymbol{x},y)\sim P_{pq}} \ell\left(\langle \boldsymbol{x}, \boldsymbol{w}_{pq}\rangle - y\right),$$

where $P_{pq}$ is the distribution from which the samples $(\boldsymbol{x}_{ipq}, y_{ipq})_{i=1}^{m_{pq}}$ are drawn from.

The next lemma relates the excess risk $L(\hat{\mathcal{W}}) - L(\mathcal{W}^*)$ with the expected dual norm $\mathbb{E}\|\mathcal{D}\|_{\star^*}$ through Rademacher complexity.

**Lemma 1.** *We assume that the output $y_{ipq}$ is bounded as $|y_{ipq}| \leq b$, and the number of samples $m_{pq} \geq m > 0$ for the observed tasks. We also assume that the loss function $\ell$ is Lipschitz continuous with the constant $\Lambda$, bounded in $[0,c]$ and $\ell(0) = 0$. Let $\mathcal{W}^*$ be any tensor such that $\|\mathcal{W}^*\|_{\star} \leq B_0$. Then with probability at least $1 - \delta$, any minimizer of (6) satisfies the following bound:*

$$L(\hat{\mathcal{W}}) - L(\mathcal{W}^*) \leq 2\Lambda\left(\frac{2B_0}{|S|}\mathbb{E}\|\mathcal{D}\|_{\star^*} + \frac{b\sqrt{\rho}}{\sqrt{|S|m}}\right) + c'\sqrt{\frac{\log(2/\delta)}{2|S|m}},$$

*where $c' = c+1$, $\|\cdot\|_{\star^*}$ is the dual norm of $\|\cdot\|_{\star}$, $\rho := \frac{1}{|S|}\sum_{(p,q)\in S} \frac{m_{pq}}{m}$. The tensor $\mathcal{D} \in \mathbb{R}^{d\times P\times Q}$ is defined as the sum $\mathcal{D} = \sum_{(p,q)\in S}\sum_{i=1}^{m_{pq}}\mathcal{Z}^{ipq}$, where $\mathcal{Z}^{ipq} \in \mathbb{R}^{d\times P\times Q}$ is defined as*

$$(p',q')\text{th fiber of } \mathcal{Z}^{ipq} = \begin{cases} \frac{1}{m_{pq}}\sigma_{ipq}\boldsymbol{x}_{ipq}, & \text{if } p=p' \text{ and } q=q', \\ 0, & \text{otherwise.} \end{cases}$$

*Here $\sigma_{ipq} \in \{-1,+1\}$ are Rademacher random variables and the expectation in the above inequality is with respect to $\sigma_{ipq}$, the random draw of tasks $S$, and the training samples $(\boldsymbol{x}_{ipq}, y_{ipq})_{i=1}^{m_{pq}}$.*

*Proof.* The proof is a standard one following the line of [5] and it is presented in Appendix A. $\quad\square$

The next theorem computes the expected dual norm $\mathbb{E}\|\mathcal{D}\|_{\star^*}$ for the three norms for tensors (the proof can be found in Appendix B).

**Theorem 1.** *We assume that $\boldsymbol{C}_{pq} := \mathbb{E}[\boldsymbol{x}_{ipq}\boldsymbol{x}_{ipq}^\top] \preceq \frac{\kappa}{d}\boldsymbol{I}_d$ and there is a constant $R > 0$ such that $\|\boldsymbol{x}_{ipq}\| \leq R$ almost surely. Let us define*

$$D_1 := d + PQ, \quad D_2 := P + dQ, \quad D_3 := Q + dP.$$

*In order to simplify the presentation, we assume that $\max_k D_k \geq 3$ and $dPQ \geq \max(d^2, P^2, Q^2)$. For the overlapped trace norm, the latent trace norm, and the scaled latent trace norm, the expectation $\mathbb{E}\|\mathcal{D}\|_{\star^*}$ can be bounded as follows:*

$$\frac{1}{|S|}\mathbb{E}\|\mathcal{D}\|_{\text{overlap}^*} \leq C\min_k\left(\sqrt{\frac{\kappa}{m|S|dPQ}D_k\log D_k} + \frac{R}{m|S|}\log D_k\right), \tag{7}$$

$$\frac{1}{|S|}\mathbb{E}\|\mathcal{D}\|_{\text{latent}^*} \leq C'\left(\sqrt{\frac{\kappa}{m|S|dPQ}\max_k(D_k\log D_k)} + \frac{R}{m|S|}\log(\max_k D_k)\right), \tag{8}$$

$$\frac{1}{|S|}\mathbb{E}\|\mathcal{D}\|_{\text{scaled}^*} \leq C''\left(\sqrt{\frac{\kappa}{m|S|}\log(\max_k D_k)} + \frac{R\sqrt{\max_k n_k}}{m|S|}\log(\max_k D_k)\right), \tag{9}$$

*where $C, C', C''$ are constants, $n_1 = d, n_2 = P$, and $n_3 = Q$. Furthermore, if $m|S| \geq R^2(\max_k n_k)\log(\max_k D_k)/\kappa$, the $O(1/m|S|)$ terms in the above inequalities can be dropped.*

Note that the assumption that the norm of $\boldsymbol{x}_{ipq}$ is bounded is natural because the target $y_{ipq}$ is also bounded. The parameter $\kappa$ in the assumption $\boldsymbol{C}_{pq} \preceq \kappa/d\boldsymbol{I}_d$ controls the amount of correlation in the data. Since $\text{Tr}(\boldsymbol{C}) = \mathbb{E}\|\boldsymbol{x}_{ipq}\|^2 \leq R^2$, we have $\kappa = O(1)$ when the features are uncorrelated; on the other hand, we have $\kappa = O(d)$, if they lie in a one dimensional subspace. The number of samples $m|S| = \tilde{O}(\max_k n_k)$ is enough to drop the $O(1/m|S|)$ term even if $\kappa = O(1)$.

Now we state the consequences of Theorem 1 for the three norms for tensors. The common assumptions are the same as in Lemma 1 and Theorem 1. We also assume $m|S| \geq R^2(\max_k n_k)\log(\max_k D_k)/\kappa$ to drop the $O(1/m|S|)$ terms. Let $\mathcal{W}^*$ be any $d \times P \times Q$ tensor with multilinear-rank $(r_1, r_2, r_3)$ and bounded element-wise as $\|\mathcal{W}^*\|_{\ell_\infty} \leq B$.

**Corollary 1** (Overlapped trace norm). *With probability at least $1 - \delta$, any minimizer of (6) with $\|\mathcal{W}\|_{\text{overlap}} \leq B\sqrt{\|\boldsymbol{r}\|_{1/2}dPQ}$ satisfies the following inequality:*

$$L(\hat{\mathcal{W}}) - L(\mathcal{W}^*) \leq c_1\Lambda B\sqrt{\frac{\kappa}{m|S|}\|\boldsymbol{r}\|_{1/2}\min_k(D_k\log D_k)} + c_2\Lambda b\sqrt{\frac{\rho}{m|S|}} + c_3\sqrt{\frac{\log(2/\delta)}{m|S|}},$$

*where $\|\boldsymbol{r}\|_{1/2} = (\sum_{k=1}^3\sqrt{r_k}/3)^2$ and $c_1, c_2, c_3$ are constants.*

Note that Tomioka et al. [23] obtained a bound that depends on $(\sum_{k=1}^3\sqrt{D_k}/3)^2$ instead of $\min(D_k\log D_k)$. Although the minimum may look better than the average, our bound has the worse constant $K = 3$ hidden in $c_1$. The $\log D_k$ factor allows us to apply the above result to the setting of tensor completion as we show below.

**Corollary 2** (Latent trace norm). *With probability at least $1 - \delta$, any minimizer of (6) with $\|\mathcal{W}\|_{\text{latent}} \leq B\sqrt{\min_k r_k dPQ}$ satisfies the following inequality:*

$$L(\hat{\mathcal{W}}) - L(\mathcal{W}^*) \leq c_1'\Lambda B\sqrt{\frac{\kappa}{m|S|}\min_k r_k \max_k(D_k\log D_k)} + c_2\Lambda b\sqrt{\frac{\rho}{m|S|}} + c_3\sqrt{\frac{\log(2/\delta)}{m|S|}},$$

*where $c_1', c_2, c_3$ are constants.*

**Corollary 3** (Scaled latent trace norm). *With probability at least $1 - \delta$, any minimizer of (6) with $\|\mathcal{W}\|_{\text{scaled}} \leq B\sqrt{\min_k(r_k/n_k)dPQ}$ satisfies the following inequality:*

$$L(\hat{\mathcal{W}}) - L(\mathcal{W}^*) \leq c_1''\Lambda B\sqrt{\frac{\kappa}{m|S|}\min_k\left(\frac{r_k}{n_k}\right)dPQ\log(\max_k D_k)} + c_2\Lambda b\sqrt{\frac{\rho}{m|S|}} + c_3\sqrt{\frac{\log(2/\delta)}{m|S|}},$$

*where $n_1 = d, n_2 = P, n_3 = Q$, and $c_1'', c_2, c_3$ are constants.*

We summarize the implications of the above corollaries for different settings in Table 2. We almost recover the settings for matrix completion [11] and multitask learning (MTL) [16]. Note that these simpler problems sometimes disguise themselves as the more general tensor completion or multilinear multitask learning problems. Therefore it is important that the new tensor based norms adapts to the simplicity of the problems in these cases.

Matrix completion is when $d = \kappa = m = r_1 = 1$, and we assume that $r_2 = r_3 = r < P, Q$. The sample complexities are the number of samples $|S|$ that we need to make the leading term in Corollaries 1, 2, and 3 equal $\epsilon$. We can see that the overlapped trace norm and the scaled latent trace norm recover the known result for matrix completion [11]. The plain latent trace norm requires $O(PQ)$ samples because it recognizes the first mode as the mode with the lowest rank 1. Although the rank $r$ of the last two modes are low *relative to their dimensions*, the latent trace norm fails to recognize this.

In multitask learning (MTL), only the first mode corresponding to features has a low rank $r$ and the other two modes have full rank. Note that a tensor *is* a matrix when its multilinear rank is full except for one mode. We also assume that all the pairs $(p,q)$ are observed ($|S| = PQ$) as in [16]. The sample complexities are defined the same way as above with respect to the number of samples $m$ because $|S|$ is fixed. The homogeneous case is when $d = P = Q$. The heterogeneous case is when $P \leq r < d$. Our bound for the overlapped trace norm is almost as good as the one in [16] but has an multiplicative $\log(PQ)$ factor (as oppose to their additive $\log(PQ)$ term) and $\|r\|_{1/2} \geq r$. Also note that the results in [16] can be applied when $d$ is much larger than $P$ and $Q$. Turning back to our bounds, both the latent trace norm and its scaled version can *perform as well as knowing the mode with the lowest rank* (the first mode) (see also [21]) when $d = P = Q$. However, when the dimensions are heterogeneous, similarly to the matrix completion case above, the plain latent trace norm fails to recognize the low-rank-ness of the first mode and

Table 2: Sample complexities of the overlapped trace norm, latent trace norm, and the scaled latent trace norm in various settings. The common factor $1/\epsilon^2$ is omitted from the sample complexities. The sample complexities are defined with respect to $|S|$ for matrix completion, $m$ for multitask learning, and $m|S|$ for tensor completion and multilinear multitask learning. In the heterogeneous cases, we assume $P \leq r < r'$. We define $\|r\|_{1/2} = (\sum_{k=1}^{3} \sqrt{r_k}/K)^2$ and $N := n_1 n_2 n_3$.

| | $(n_1, n_2, n_3)$ | $(r_1, r_2, r_3)$ | $(\kappa, B, |S|)$ | Sample complexities (per $1/\epsilon^2$) | | |
| --- | --- | --- | --- | --- | --- | --- |
| | | | | Overlap | Latent | Scaled |
| Matrix completion [11] | 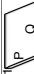 | $(1, r, r)$ | $(1, 1, |S|)$ | $\|r\|_{1/2}(P+Q)\log(P+Q)$ | $PQ\log(PQ)$ | $r(P+Q)\log(PQ)$ |
| MTL [16] (homogeneous case) |  | $(r, d, d)$ | $(d, 1/\sqrt{d}, d^2)$ | $\|r\|_{1/2}\log(d^2)$ | $r\log(d^2)$ | $r\log(d^2)$ |
| MTL (heterogeneous case) |  | $(r, P, r')$ | $(d, 1/\sqrt{d}, PQ)$ | $\|r\|_{1/2}\log(PQ)$ | $d\log(dQ)$ | $r\log(dQ)$ |
| MLMTL [17] (homogeneous case) |  | $(r_1, r_2, r_3)$ | $(\kappa, 1, |S|)$ | $\kappa\|r\|_{1/2}d^2\log(d^2)$ | $\kappa(\min_k r_k)d^2\log(d^2)$ | $\kappa(\min_k r_k)d^2\log(d^2)$ |
| MLMTL [17] (heterogeneous case) |  | $(r, P, r')$ | $(\kappa, 1, |S|)$ | $\kappa\|r\|_{1/2}PQ\log(PQ)$ | $\kappa dPQ\log(dQ)$ | $\kappa\min(r\cdot PQ, dPr')\log(dQ)$ |
| Tensor completion |  | $(r_1, r_2, r_3)$ | $(1, 1, |S|)$ | $\|r\|_{1/2}\min_k(D_k\log D_k)$ | $\min_k r_k \max_k(D_k\log D_k)$ | $\min_k(\frac{r_k}{n_k})N\log(\max_k D_k)$ |

requires $O(d)$ samples, because the second mode has the lowest rank $P$.

In multilinear multitask learning (MLMTL) [17], any mode could possibly be low rank but it is a priori unknown. The sample complexities are defined the same way as above with respect to $m|S|$. The homogeneous case is when $d = P = Q$. The heterogeneous case is when the first mode or the third mode is low rank but $P \leq r < d$. Similarly to the above two settings, the overlapped trace norm has a mild dependence on the dimensions but a higher dependence on the rank $\|r\|_{1/2} \geq r$. The latent trace norm performs as well as knowing the mode that has the lowest rank in the homogeneous case. However, it fails to recognize the mode with the lowest rank relative to its dimension. The scaled latent trace norm does this and although it has a higher logarithmic dependence, it is competitive in both cases.

Finally, our bounds also hold for tensor completion. Although Tomioka et al. [22, 23] studied tensor completion algorithms, their analysis assumed that the inputs $x_{ipq}$ are drawn from a Gaussian distribution, which does not hold for tensor completion. Note that in our setting $x_{ipq}$ can be an indicator vector that has one in the $j$th position uniformly over $1, \ldots, d$. In this case, $\kappa = 1$. The sample complexities of different norms with respect to $m|S|$ is shown in the last row of Table 2. The sample complexity for the overlapped trace norm is the same as the one in [23] with a logarithmic factor. The sample complexities for the latent and scaled latent trace norms are new. Again we can see that while the latent trace norm recognize the mode with the lowest rank, the scaled latent trace norm is able to recognize the mode with the lowest rank relative to its dimension.

## 4 Experiments

We conducted several experiments to evaluate performances of tensor based multitask learning setting we have discussed in Section 3. In Section 4.1, we discuss simulation we conducted using synthetic data sets. In Sections 4.2 and 4.3, we discuss experiments on two real world data sets, namely the Restaurant data set [26] and School Effectiveness data set [3, 4]. Both of our real world data sets have heterogeneous dimensions (see Figure 2) and it is a priori unclear across which mode has the most amount of information sharing.

### 4.1 Synthetic data sets

The true $d \times P \times Q$ tensor $\mathcal{W}^*$ was generated by first sampling a $r_1 \times r_2 \times r_3$ core tensor and then multiplying random orthonormal matrix to each of its modes. For each task $(p, q) \in [P] \times [Q]$, we generated training set of $m$ vectors $(x_{ipq}, y_{ipq})_{i=1}^m$ by first sampling $x_{ipq}$ from the standard normal distribution and then computing $y_{ipq} = \langle x_{ipq}, w_{pq} \rangle + \nu_i$, where $\nu_i$ was drawn from a zero-mean normal distribution with variance $0.1$. We used the penalty formulation of (6) with the squared loss and selected the regularization parameter $\lambda$ using two-fold cross validation on the training set from the range $0.01$ to $10$ with the interval $0.1$.

In addition to the three norms for tensors we discussed in the previous section, we evaluated the matrix-based multitask learning approaches that penalizes the trace norm of the unfolding of $\mathcal{W}$ at specific modes. The conventional convex multitask learning [2, 3, 16] corresponds to one of these approaches that penalizes the trace norm of the first unfolding $\|W_{(1)}\|_{\mathrm{tr}}$. The convex MLMTL in [17] corresponds to the overlapped trace norm.

In the first experiment, we chose $d = P = Q = 10$ and $r_1 = r_2 = r_3 = 3$. Therefore, both the dimensions and the multilinear rank are homogeneous. The result is shown in Figure 1(a). The overlapped trace norm performed the best, the matrix-based approaches performed next, and the latent trace norm and the scaled latent trace norm were the worst. The scaling of the latent trace norm had no effect because the dimensions were homogeneous. Since the sample complexities for all the methods were the same in this setting (see Table 2), the difference in the performances could be explained by a constant factor $K(= 3)$ that is not shown in the sample complexities.

In the second experiment, we chose the dimensions to be homogeneous as $d = P = Q = 10$, but $(r_1, r_2, r_3) = (3, 6, 8)$. The result is shown in Figure 1(b). In this setting, the (scaled) latent trace norm and the mode-1 regularization performed the best. The lower the rank of the corresponding mode, the lower were the error of the matrix-based MTL approaches. The overlapped trace norm was somewhat in the middle of the three matrix-based approaches.

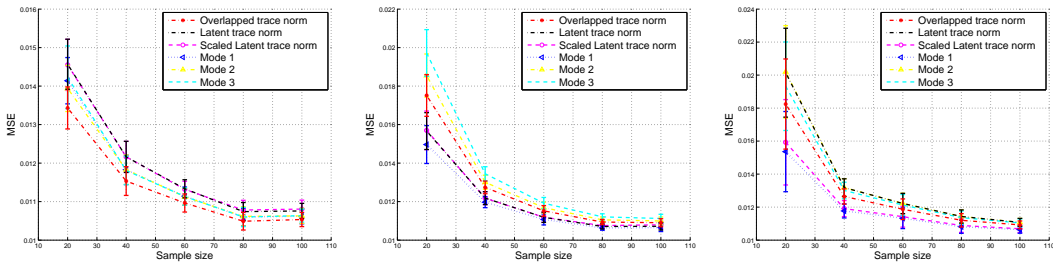

(a) Synthetic experiment for the case when both the dimensions and the ranks are homogeneous. The true tensor is $10 \times 10 \times 10$ with multilinear rank $(3, 3, 3)$.

(b) Synthetic experiment for the case when the dimensions are homogeneous but the ranks are heterogeneous. The true tensor is $10 \times 10 \times 10$ with multilinear rank $(3, 6, 8)$.

(c) Synthetic experiment for the case when both the dimensions and the ranks are heterogeneous. The true tensor is $10 \times 3 \times 10$ with multilinear rank $(3, 3, 8)$.

Figure 1: Results for the synthetic data sets.

In the last experiment, we chose both the dimensions and the multilinear rank to be heterogeneous as $(d, P, Q) = (10, 3, 10)$ and $(r_1, r_2, r_3) = (3, 3, 8)$. The result is shown in Figure 1(c). Clearly the first mode had the lowest rank relative to its dimension. However, the latent trace norm recognizes the second mode as the mode with the lowest rank and performed similarly to the mode-2 regularization. The overlapped trace norm performed better but it was worse than the mode-1 regularization. The scaled latent trace norm performed comparably to the mode-1 regularization.

## 4.2 Restaurant data set

The Restaurant data set contains data for a recommendation system for restaurants where different customers have given ratings to different aspects of each restaurant. Following the same approach as in [17] we modelled the problem as a MLMTL problem with $d = 45$ features, $P = 3$ aspects, and $Q = 138$ customers.

The total number of instances for all the tasks were $3483$ and we randomly selected training set of sizes 400, 800, 1200, 1600, 2000, 2400, and 2800. When the size was small many tasks contained no training example. We also selected 250 instances as the validation set and the rest was used as the test set. The regularization parameter for each norm was selected by minimizing the mean squared error on the validation set from the candidate values in the interval $[50, 1000]$ for the overlapped, $[0.5, 40]$ for the latent, $[6000, 20000]$ for the scaled latent norms, respectively.

We also evaluated matrix-based MTL approaches on different modes and ridge regression (Frobenius norm regularization; abbreviated as RR) as baselines. The convex MLMTL in [17] corresponds to the overlapped trace norm.

The result is shown in Figure 2(a). We found the multilinear rank of the solution obtained by the overlapped trace norm to be typically $(1, 3, 3)$. This was consistent with the fact that the performances of the mode-1 regularization and the ridge regression were equal. In other words, the effective dimension of the first mode (features) was one instead of 45. The latent trace norm recognized the first mode as the mode with the lowest rank and it failed to take advantage of the low-rank-ness of the second and the third modes. The scaled latent trace norm was able to perform the best matching with the performances of mode-2 and mode-3 regularization. When the number of samples was above 2400, the latent trace norm caught up with other methods, probably because the effective dimension became higher in this regime.

## 4.3 School data set

The data set comes from the inner London Education Authority (ILEA) consisting of examination records from 15362 students at 139 schools in years 1985, 1986, and 1987. We followed [4] for the preprocessing of categorical attributes and obtained 24 features. Previously Argyriou et al. [3] modeled this data set as a $27 \times 139$ matrix-based MTL problem in which the year was modeled as a

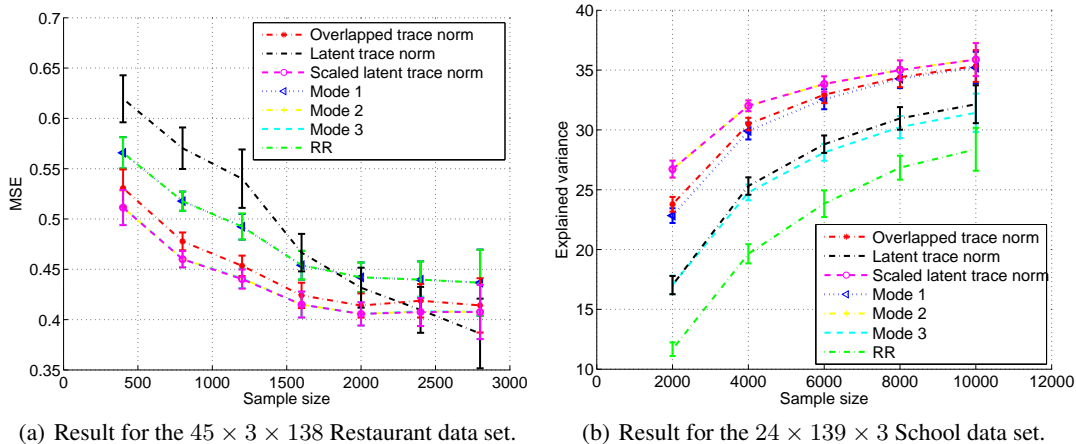

(a) Result for the $45 \times 3 \times 138$ Restaurant data set.

(b) Result for the $24 \times 139 \times 3$ School data set.

Figure 2: Results for the real world data sets.

trinomial attribute. Instead here we model this data set as a $24 \times 139 \times 3$ MLMTL problem in which the third mode corresponds to the year. Following earlier papers, [3, 4], we used the percentage of explained variance, defined as $100 \cdot (1 - (\text{test MSE})/(\text{variance of } y))$, as the evaluation metric.

The results are shown in Figure 2(b). First, ridge regression performed the worst because it was not able to take advantage of the low-rank-ness of any mode. Second, the plain latent trace norm performed similarly to the mode-3 regularization probably because the dimension 3 was lower than the rank of the other two modes. Clearly the scaled latent trace norm performed the best matching with the performance of the mode-2 regularization; probably the second mode had the most redundancy. The performance of the overlapped trace norm was comparable or slightly better than the mode-1 regularization. The percentage of the explained variance of the latent trace norm exceeds 30 % around sample size 4000 (around 30 samples per school), which is higher than the Hierarchical Bayes [4] (around 29.5 %) and matrix-based MTL [3] (around 26.7 %) that used around 80 samples per school.

## 5 Discussion

Using tensors for modeling multitask learning [17, 19] is a promising direction that allows us to take advantage of similarity of tasks in multiple dimensions and even make prediction for a task with no training example. However, having multiple modes, we would have to face with more hyperparameters to choose in the conventional nonconvex tensor decomposition framework. Convex relaxation of tensor multilinear rank allows us to side-step this issue. In fact, we have shown that the sample complexity of the latent trace norm is as good as knowing the mode with the lowest rank. This is consistent with the analysis of [21] in the tensor denoising setting (see Table 1).

In the setting of tensor-based MTL, however, the notion of mode with the lowest rank may be vacuous because some modes may have very low dimension. In fact, the sample complexity of the latent trace norm can be as bad as not using any low-rank-ness at all if there is a mode with dimension lower than the rank of the other modes. The scaled latent trace norm we proposed in this paper recognizes the mode with the lowest rank *relative to its dimension* and lead to the competitive sample complexities in various settings we have shown in Table 2.

**Acknowledgment**: MS acknowledges support from the JST CREST program.

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
