[Supplementary Material]

# A   Proof of Lemma 1

*Proof.*  The proof follows a standard argument, which can be found in Bartlett and Mendelson [5, Theorem 8].

$$L(\hat{\mathcal{W}}) - L(\mathcal{W}^*) \leq \left( L(\hat{\mathcal{W}}) - \hat{L}(\hat{\mathcal{W}}) \right) + \left( \hat{L}(\hat{\mathcal{W}}) - \hat{L}(\mathcal{W}^*) \right) + \left( \hat{L}(\mathcal{W}^*) - L(\mathcal{W}^*) \right)$$

$$\leq \sup_{\|\|\mathcal{W}\|\|_\star \leq B_0} \left( L(\mathcal{W}) - \hat{L}(\mathcal{W}) \right) + \sqrt{\frac{\log(2/\delta)}{2\rho|S|m}} \quad \text{(w/ probability at least } 1 - \delta/2\text{)}$$

$$\leq R(\ell \circ \mathcal{L}_{B_0}) + \left( c + \frac{1}{\sqrt{\rho}} \right) \sqrt{\frac{\log(2/\delta)}{2|S|m}} \quad \text{(w/ probability at least } 1 - \delta\text{)},$$

where

$$R(\ell \circ \mathcal{L}_{B_0}) := \mathbb{E} \sup_{\|\|\mathcal{W}\|\|_\star \leq B_0} \frac{2}{|S|} \sum_{(p,q) \in S} \frac{1}{m_{pq}} \sum_{i=1}^{m_{pq}} \sigma_{ipq} \ell \left( \langle \boldsymbol{x}_{ipq}, \boldsymbol{w}_{pq} \rangle - y_{ipq} \right).$$

In the third line, we used McDiarmid's inequality and introduced Rademacher random variables $\sigma_{ipq} \in \{-1, +1\}$; the expectation is over both the Rademacher random variables and the training samples $(\boldsymbol{x}_{ipq}, y_{ipq})$. Using the fact that $c + 1/\sqrt{\rho} \leq c + 1 =: c'$, the last term can be upper bounded by the last term in the statement.

We further analyze the first term. Using the Lipschitz continuity of $\ell$ and the bound on $|y_{ipq}|$, we have

$$R(\ell \circ \mathcal{L}_{B_0}) \leq 2\Lambda \left( R(\mathcal{L}_{B_0}) + \frac{b\sqrt{\sum_{(p,q) \in S} m_{p,q}}}{|S|m} \right),$$

where

$$R(\mathcal{L}_B) = \frac{2}{|S|} \mathbb{E} \sup_{\|\|\mathcal{W}\|\|_\star \leq B_0} \sum_{(p,q) \in S} \frac{1}{m_{pq}} \sum_{i=1}^{m_{pq}} \sigma_{ipq} \langle \boldsymbol{x}_{ipq}, \boldsymbol{w}_{pq} \rangle.$$

Finally, using the definition of $\mathcal{D}$ and Hölder's inequality, we have

$$R(\mathcal{L}_{B_0}) \leq \frac{2B_0}{|S|} \mathbb{E} \|\|\mathcal{D}\|\|_{\star^*},$$

which concludes the proof. $\qquad\qquad\qquad\qquad\qquad\qquad\qquad\qquad\qquad\qquad\qquad\qquad\qquad\qquad\qquad\quad\square$

# B   Proof of Theorem 1

**Proof of inequality** (7)**:**   From Tomioka et al. [23, Lemma 1], we have

$$\|\|\mathcal{D}\|\|_{\text{overlap}^*} = \inf_{\mathcal{D}^{(1)} + \mathcal{D}^{(2)} + \mathcal{D}^{(3)} = \mathcal{D}} \max_k \|\boldsymbol{D}_{(k)}^{(k)}\|_{\text{op}},$$

where the infimum is over three tensors $\mathcal{D}^{(1)}$, $\mathcal{D}^{(2)}$, and $\mathcal{D}^{(3)}$ that sum to the original tensor $\mathcal{D}$, and $\| \cdot \|_{\text{op}}$ is the operator norm (maximal singular value). Since we can take any $\mathcal{D}^{(k)}$ to equal $\mathcal{D}$, the norm can be upper bounded as follows:

$$\|\|\mathcal{D}\|\|_{\text{overlap}^*} \leq \min_k \|\boldsymbol{D}_{(k)}\|_{\text{op}}.$$

Since the expectation of minimum over $k$ can be upper bounded by the minimum of expectations, we have

$$\mathbb{E} \|\|\mathcal{D}\|\|_{\text{overlap}^*} \leq \mathbb{E} \min_k \|\boldsymbol{D}_{(k)}\|_{\text{op}} \leq \min_k \mathbb{E} \|\boldsymbol{D}_{(k)}\|_{\text{op}}.$$

Now we upper bound each expectation using Theorem 6.1 in Tropp [24, see also Remarks 6.3 and 6.5], which states that

$$\Pr \left\{ \|\boldsymbol{D}_{(k)}\|_{\text{op}} \geq t \right\} \leq \begin{cases} D_k \exp(-3t^2/8\sigma_k^2), & \text{for } t \leq \sigma_k^2/R_k, \\ D_k \exp(-3t/8R_k), & \text{for } t \geq \sigma_k^2/R_k, \end{cases} \qquad (10)$$

and

$$\mathbb{E}\|\boldsymbol{D}_{(k)}\|_{\mathrm{op}} \leq C(\sigma_k \sqrt{\log D_k} + R_k \log D_k),\tag{11}$$

where $C$ is an absolute constant, and

$$\sigma_k^2 := \max \left( \left\| \sum_{(p,q)\in S} \sum_{i=1}^{m_{pq}} \mathbb{E}\left[ \boldsymbol{Z}_{(k)}^{ipq} \left( \boldsymbol{Z}_{(k)}^{ipq} \right)^\top \right] \right\|_{\mathrm{op}}, \left\| \sum_{(p,q)\in S} \sum_{i=1}^{m_{pq}} \mathbb{E}\left[ \left( \boldsymbol{Z}_{(k)}^{ipq} \right)^\top \boldsymbol{Z}_{(k)}^{ipq} \right] \right\|_{\mathrm{op}} \right),$$

$$R_k \geq \left\| \boldsymbol{Z}_{(k)}^{ipq} \right\|_{\mathrm{op}} \quad \text{(almost surely)}.$$

Due to our assumption $\|\boldsymbol{x}_{ipq}\| \leq R$, we can take $R_k = R/m$. Thus the remaining task is to compute $\sigma_k^2$ for $k = 1, 2, 3$.

First for $k = 1$, the unfolding $\boldsymbol{Z}_{(1)}^{ipq}$ is a $d \times PQ$ matrix that contains $\sigma_{ipq}\boldsymbol{x}_{ipq}/m_{pq}$ in the column specified by $(p, q)$. Therefore, using $m_{pq} \geq m$ and $\|\boldsymbol{C}_{pq}\| \leq \kappa/d$, we obtain

$$\sum_{i=1}^{m_{pq}} \mathbb{E}\left[ \boldsymbol{Z}_{(1)}^{ipq} \left( \boldsymbol{Z}_{(1)}^{ipq} \right)^\top \right] = \frac{1}{m_{pq}} \boldsymbol{C}_{pq} \preceq \frac{\kappa}{md} \boldsymbol{I}_d,$$

from which we have

$$\left\| \sum_{(p,q)\in S} \sum_{i=1}^{m_{pq}} \mathbb{E}\left[ \boldsymbol{Z}_{(1)}^{ipq} \left( \boldsymbol{Z}_{(1)}^{ipq} \right)^\top \right] \right\|_{\mathrm{op}} \leq \frac{\kappa|S|}{md}.\tag{12}$$

Similarly, since the choice of $(p, q)$ is uniform over $[P] \times [Q]$, we have

$$\sum_{i=1}^{m_{pq}} \mathbb{E}\left[ \left( \boldsymbol{Z}_{(1)}^{ipq} \right)^\top \boldsymbol{Z}_{(1)}^{ipq} \right] = \frac{1}{PQ}\mathrm{diag}\left( \frac{\mathrm{Tr}\boldsymbol{C}_{pq}}{m_{pq}} \right) \preceq \frac{\kappa}{mPQ} \boldsymbol{I}_{PQ},$$

from which we have

$$\left\| \sum_{(p,q)\in S} \sum_{i=1}^{m_{pq}} \mathbb{E}\left[ \left( \boldsymbol{Z}_{(1)}^{ipq} \right)^\top \boldsymbol{Z}_{(1)}^{ipq} \right] \right\|_{\mathrm{op}} \leq \frac{\kappa|S|}{mPQ}.\tag{13}$$

Substituting inequalities (12) and (13) into (11), we have

$$\mathbb{E}\|\boldsymbol{D}_{(1)}\|_{\mathrm{op}} \leq C\left( \sqrt{\frac{\kappa|S|}{mdPQ}D_1 \log D_1} + \frac{R}{m}\log D_1 \right).$$

Following a similar line of argument, we have

$$\mathbb{E}\|\boldsymbol{D}_{(2)}\|_{\mathrm{op}} \leq C\left( \sqrt{\frac{\kappa|S|}{mdPQ}D_2 \log D_2} + \frac{R}{m}\log D_2, \right),$$

$$\mathbb{E}\|\boldsymbol{D}_{(3)}\|_{\mathrm{op}} \leq C\left( \sqrt{\frac{\kappa|S|}{mdPQ}D_3 \log D_3} + \frac{R}{m}\log D_3, \right).$$

Taking the minimum over $k$ and dividing by $|S|$, we obtain inequality (7). $\qquad\square$

**Proof of inequality (8):** From Tomioka et al. [21, Lemma 1], we know that

$$\|\mathcal{D}\|_{\mathrm{latent}*} = \max_k \|\boldsymbol{D}_{(k)}\|_{\mathrm{op}}.$$

Combining inequality (10) with a union bound, we have

$$\Pr\left\{ \|\mathcal{D}\|_{\mathrm{latent}*} \geq t \right\} \leq 3(\max_k D_k) \max\left( \exp\left( -\frac{3t^2}{8\max_k \sigma_k^2} \right), \exp\left( -\frac{3t}{8\max_k R_k} \right) \right),$$

from which we have

$$\mathbb{E}\|\mathcal{D}\|_{\mathrm{latent}*} \leq C\left( \max_k \sigma_k \sqrt{\log(\max_k D_k) + \log 3} + \max_k R_k(\log(\max_k D_k) + \log 3) \right)\tag{14}$$

$$\leq C'\left( \max_k \sigma_k \sqrt{\log(\max_k D_k)} + \frac{R}{m}\log(\max_k D_k) \right).$$

Here we used $R_k = R/m$ and the simplifying assumption that $\max_k D_k \geq 3$ in the second inequality. Finally, using $\sigma_k \leq \sqrt{\kappa|S|D_k/(mdPQ)}$ as in the proof of inequality (7), we obtain inequality (8).

**Proof of inequality** (9): Following the proof of [21, Lemma 1], we have

$$\|\mathcal{D}\|_{\text{scaled}*} = \max_k \sqrt{n_k}\|\boldsymbol{D}_{(k)}\|_{\text{tr}},$$

where $n_1 = d$, $n_2 = P$, and $n_3 = Q$. Thus, replacing $\sigma_k$ and $R_k$ with $\sqrt{n_k}\sigma_k$ and $\sqrt{n_k}R/m$ in inequality (14), respectively, we have

$$\mathbb{E}\|\mathcal{D}\|_{\text{scaled}*} \leq C'\left(\max_k(\sqrt{n_k}\sigma_k)\sqrt{\log(\max_k D_k)} + \frac{R\sqrt{\max_k n_k}}{m}\log(\max_k D_k)\right).$$

Finally, since $n_k D_k = n_k^2 + dPQ \leq 2dPQ$, we have

$$\sqrt{n_k}\sigma_k \leq \sqrt{\frac{\kappa|S|n_k D_k}{mdPQ}} \leq \sqrt{\frac{2\kappa|S|}{m}},$$

which gives inequality (9).

The last claim of the theorem is true, because $m|S| \geq R^2(\max_k n_k)(\log_k D_k)/\kappa$ implies

$$m|S| \geq \frac{R^2}{\kappa}\frac{dPQ}{n_k^2 + dPQ}n_k \log D_k = \frac{R^2}{\kappa}\frac{dPQ}{D_k}\log D_k,$$

which gives

$$\sqrt{\frac{\kappa}{m|S|dPQ}}D_k \log D_k \geq \frac{R}{m|S|}\log D_k.$$