[Reviews · NeurIPS 2014]

Submitted by Assigned_Reviewer_9

In this paper, the authors study the problem of learning a tensor for the purpose of linear multi-task learning. The authors propose a new weighted version of a previously proposed tensor norm (called "latent trace norm") and show that the introduced rescaling yields better bounds on the excess risk as well as improved recovery performance on some datasets. The paper is well written and organized, and the proposed rescaling can potentially have a significant impact in practice, although a more extensive experimental evaluation would have been desirable. The technical results seem to be appropriate and correctly proven. Literature coverage seems to be sufficient. It may be worth noticing that all the tensor norms studied in this paper can be also seen as particular cases of a more general class of tensor penalties introduced in the recent paper
A. Argyriou, F. Dinuzzo. A Unifying View of Representer Theorems. ICML 2014

Summary: The paper carries out a study of some recently proposed tensor norms, and a new weighted version of one of them, illustrating the advantages of such reweighting both theoretically and empirically. The paper is well-written, organized, and of potential significance.

Submitted by Assigned_Reviewer_41

In this paper, the authors proposed a new norm, called scaled latent trace norm, to relax the convex condition for the tensor multilinear rank. Both theoretical and experimental results are presented to show that the advantage of the scaled latent trace norm especially when the multitask learning task has inhomogeneous dimensions and there is no priori knowledge about which mode is low rank.

Strengths about this paper:
1) This paper is well written.
2) The authors develop nice theorems that show the upper bound of the error between the empirical risk and the true risk in different scenarios in which overlapped trace norm, latent norm, and scaled latent norm are involved respectively.
3) The authors also list all sample complexity for matrix completion, multitask learning, and multilinear multitask learning to compare the results.
4) The authors provide corresponding experimental results to show that the scaled latent norm performs better when multitask learning involves inhomogeneous dimensions.

Some aspects to clarify/improve:
1) Latent trace norm is studied extensively in R. Tomioka and T.suzuki (2013), and the contribution of this paper is to develop "scaled" latent trace norm. The difference is not that big.
2) From the results (Table 2), the sample complexity of the scaled latent trace norm is a little better than that of the latent trace norm. In tensor completion, it's hard to tell whether scaled latent trace norm is always better than that of the latent trace norm.
3) The upper bound of all scenarios involved in three norms are all proportional to the number of training samples to the power of negative 1/2, which makes them not essentially distinguished.
Summary: In general a good paper with nice theoretical results.

Submitted by Assigned_Reviewer_42

After reading the authors feedback, I think I have misunderstood the "mode" in the experiment, which is exactly what I was requested. I have modified my score correspondingly.

===

In this paper the authors studied an approach for multi-task learning, which represent the (linear) models of the tasks into a tensor, and enforce a low-rank structure in the tensor to model the task relatedness, in which the knowledge is transferred among the tasks. The existing approaches such as latent trace norm failed to address the problem where the task dimensions are heterogeneous, and this paper proposed an improve version of latent trace norm, which normalizes along each dimension. The authors derived error bounds for the two existing tensor norms and the proposed one, relating them to the expected dual norm and showing the expected dual norm for the three norms. It is really nice to have a unified comparison over different norms as shown in the paper. Finally, the authors showed experimental results on synthetic data as well as two real-world data. Overall, this is a good paper with interesting theoretical analysis, and below are my comments of the paper.

My main concern of this paper is the usefulness of the tensor multi-task learning. Indeed we can have some “features X aspects X customers” model to explore the relatedness, and the question is that: is it necessary to go beyond two-dimensional case. For any tensor model, we can collapse the model into a flat “features X task” matrix and enforce a low rank to enforce the task relatedness. As one may argue this cannot capture some part of the information as did in tensor formulations, it rare to see how tensor can indeed increase the performance. Even if it helps a little bit, will people pay a large amount of additional time cost for such increase in performance? The lacking of the comparison to simple matrix trace norm in the experiments somehow exaggerated my concern.

The core novelty of this paper is the scaled version of a traditional latent trace norm, by adding a scaling factor to the unfolding along each mode. This suggests that we should treat each task (and the unfolding) differently due to some reasons. The idea is somehow related to another paper trying to address a similar problem, the “Multivariate Regression with Calibration” by Liu et. al. The authors may want to see if this paper can be related to that one. It is interesting if the authors can show some connection between the two.

What is the algorithm used in this paper to solve the formulation with the new norm? Is it the same as the one used in the original latent tensor norm? What is the complexity of algorithm? In the case where scaled latent trace norm performs better than the original version, how much time does it take to converge?

As mentioned before, for multi-task learning, everything use tensor can also be used by flat trace norm. The authors should at least compare to these types of “flat” multi-task learning methods, e.g., the trace norm, in terms of performance and efficiency. Also, it is weird to see that the authors use different evaluation (explained variance) metric for the school data. There are many multi-task learning papers that report MSE/NMSE for school data. The authors can use MSE on school data as well to make the paper consistent.

Some minor comments:
1. Notation “W_{(k)}^{(k)}” in (2) and later on is very confusing. It has not been defined before its use.
2. Typo “Turing”=>“Turning” in page 5, 10 lines from the bottom.
Summary: In the paper the authors proposed a scaled latent trace norm for multi-task learning and provided both theoretical analysis and some empirical evaluation. The paper looks good and I have some concerns on the impact of this paper.
Author Feedback
Author rebuttal: Response to Assigned_Reviewer_41:
Regarding the improvement compared to latent trace norm in Table 2: There was an error in Table 2 regarding the sample complexity of the scaled latent trace norm for MLMTL (inhomogeneous case). It should have been kappa rPQ log(dQ) instead of kappa rPd log(dQ). Thus the scaled latent trace norm is better than the plain latent trace norm by the factor r/d. We were implicitly assuming d=Q. We will make this clear in the final version.

Regarding the sample complexity for tensor completion: We would like to clarify that the scaled version is always better than the original latent trace norm. This is because
max_k D_k = > max_k (N/n_k) = N/(min_k n_k)
min_k r_k = min_k (r_k/n_k * n_k) = > min_k (r_k/n_k) min_k n_k
Thus
(min_k r_k) * (max_k D_k) = > min_k (r_k/n_k) * N.

We agree that our current bound is of O(n^{-1/2}). An advantage of the Rademacher-complexity-based technique used in this paper is its wide applicability; it applies to all three norms in various settings listed in Table 2. The technique used to derive the fast rate in Tomioka et al. (2011) applies only to the overlapped trace norm. The technique used in Tomioka and Suzuki (2013) only applies to tensor denoising (see Table 1). We would like to point out that there is a hybrid result that behaves as O(n^{-1}) when the approximation error is zero (Srebro, Sridharan, Tewari (2010)), which could possibly be applied to our setting.

Response to Assigned_Reviewer_42:
Methods called Mode 1, Mode 2, and Mode 3 use the matrix-based trace norm. In particular, Mode 1 corresponds to the flat features x tasks unfolding, which you requested.

Thank you for pointing us to Liu et al.'s paper. It seems possible to combine our higher-order modeling with their L2,1 loss function.

The algorithm used for the proposed scaled latent trace norm is basically the same dual-based ADMM as in Tomioka, Hayashi, Kashima (2011). The algorithm scales as O(m^3PQ + NK), where m is the number of samples per task, PQ is the total number of tasks, N=dPQ, and K=3 is the order of the tensor.

Response to Assigned_Reviewer_9
We will include a more through empirical evaluation in the final version. Thank you for pointing us to Argyriou & Dinuzzo's paper.